# A Comparative Study on Inhibition of Breast Cancer Cells and Tumors in Mice by Carotenoid Extract and Nanoemulsion Prepared from Sweet Potato (*Ipomoea batatas* L.) Peel

**DOI:** 10.3390/pharmaceutics14050980

**Published:** 2022-05-02

**Authors:** Hsin-Yen Hsu, Bing-Huei Chen

**Affiliations:** 1Department of Food Science, Fu Jen Catholic University, New Taipei City 24205, Taiwan; luckypinocat@gmail.com; 2Department of Nutrition, China Medical University, Taichung 40402, Taiwan

**Keywords:** sweet potato peel, carotenoid nanoemulsion, breast cancer cell, mice tumor

## Abstract

The objectives of this study were to determine carotenoid composition in sweet potato (TNG66) peel and prepare carotenoid nanoemulsion to study its inhibition effect on breast cancer cells MCF-7 and tumors in mice. Results showed that a total of 10 carotenoids were separated within 30 min by employing a YMC C30 column and a gradient mobile phase of methanol/acetonitrile/water (74:14:12, *v/v/v*) and dichloromethane (B) with a flow rate of 1 mL/min, column temperature of 25 °C, and detection wavelength of 450 nm. Following quantitation, all-trans-β-carotene was present in the highest amount (663.8 μg/g). The method validation data demonstrated a high accuracy and precision of this method. The carotenoid nanoemulsion was prepared by mixing an appropriate proportion of carotenoid extract, Tween 80, PEG 400, soybean oil and deionized water with the mean particle size being 15.7 nm (transmission electron microscope (TEM)), polydispersity index 0.238, encapsulation efficiency 97% and zeta potential −69.8 mV. A high stability of carotenoid nanoemulsion was shown over a 90-day storage period at 25 °C and during heating at 100 °C for 2 h. The release percentage of total carotenoids from carotenoid nanoemulsion under gastric and intestinal condition was 18.3% and 49.1%, respectively. An antiproliferation study revealed that carotenoid nanoemulsion was more effective than carotenoid extract in inhibiting the growth of human breast cancer cells MCF-7. Following treatments of paclitaxel (10 μg/mL), carotenoid nanoemulsion (20 and 10 μg/mL) and carotenoid extract (20 and 10 μg/mL), the tumor weight of mice respectively decreased by 77.4, 56.2, 40.3, 36.1 and 18.7%, as well as tumor volume of mice by 75.4, 65.0, 49.7, 46.7 and 26.5%. Also, both carotenoid extract and nanoemulsion could reduce the levels of epidermal growth factor (EGF) and (vascular endothelial growth factor (VEGF) in serum, with the latter being more effective. This finding suggested that carotenoid nanoemulsion was more effective than carotenoid extract in inhibiting tumor growth in mice.

## 1. Introduction

Sweet potato (*Ipomoea batatas* (L.) Lam), a vital food crop in the world, has received great attention over the years due to its reported beneficial effects on human health. In Taiwan, sweet potato is the second largest food crop with many cultivars, as the environmental condition is appropriate for growth of sweet potato in all seasons in Taiwan [1]. Most importantly, many studies have demonstrated that sweet potato roots are rich in nutrients such as carbohydrate, dietary fiber, vitamins and minerals as well as phytochemicals such as carotenoids, phenolic acids, flavonoids and anthocyanins [2,3].

Carotenoids, a group of lipid-soluble compounds with color ranging from yellow to red, are widely distributed in nature, especially green plants. They are divided into hydrocarbon carotenes and oxygenated derivatives named xanthophylls. Like sweet potato, carotenoids have been demonstrated to possess important biological activities such as anti-cancer, anti-inflammation, protection of cardiovascular disease and enhancement of immunity [4,5]. In a study dealing with the analysis of phytochemicals in sweet potato grown in Taiwan, Chen [1] reported that the total amount of carotenoids in sweet potato peel and pulp was similar with β-carotene being the major carotenoids present at a level ranging from 271.80 ± 6.68 to 450.88 ± 8.25 μg/g DW. However, the variety and amount of individual carotenoids present in sweet potato peel still remains unknown. As sweet potato peel is often discarded as waste during sweet potato processing, it would be a great advantage to the health food industry if the carotenoids can be extracted from sweet potato peel and processed into health-promoting functional foods.

Nevertheless, the unstable nature of carotenoids has limited its application to the health food industry because of the presence of long-chain conjugated carbon-carbon double bonds. To remedy this problem, many nanotechniques have been developed to encapsulate carotenoids to enhance its stability both in vitro and in vivo. Of the various nanotechniques, both nanoemulsion and microemulsion are used quite often due to ease of preparation with the size of the former ranging from 10–100 nm and the latter from 2–100 nm [6]. Specifically, both nanoemulsion and microemulsion are composed of oil, water, surfactant or co-surfactant in an appropriate proportion with microemulsion being more kinetically stable than nanoemulsion [6]. Nevertheless, a much smaller nanoparticle size (<10 nm) was reported to be rapidly eliminated by kidney in vivo [7]. Some other studies also suggested that nanoparticles with size from 10–60 nm were the most efficient for cell uptake in vivo [8], while the cellular binding probability for 15 nm nanoparticles was shown to be 13-fold higher than 150 nm nanoparticle [9]. This outcome clearly revealed that the size of nanoemulsion or microemulsion has to be carefully controlled to obtain the maximum anti-cancer activity.

According to a report by Ministry of Health and Welfare, Taiwan (MOHW) [10], malignant tumor was ranked as the first of ten leading causes of death, in which the mortality rate of female breast cancer ranked as fourth. The breast cancer therapy often includes operation, radiation, chemotherapy, hormone block and targeted therapy. However, the drug treatments may cause severe side effects. Thus, it is desirable to reduce the side effects by incorporating drugs into the nanoemulsion or microemulsion while minimizing dose intake.

Based on a previous study conducted by Chen [1], of the nine cultivars of sweet potatoes in Taiwan investigated Tainung 66 (TNG66) was reported to possess the highest amount of phytochemicals such as carotenoids and phenolic acids. However, the presence of carotenoids in TNG66 peel remains unexplored. Thus, in this study the sweet potato (TNG66) peel was selected as a raw material to prepare the carotenoid nanoemulsion and study its inhibition effect on breast cancer cells MCF-7 and tumors in mice.

## 2. Materials and Methods

### 2.1. Materials

Sweet potato peel (TNG 66) was provided by a local sweet potato processing plant located at New Taipei City, Taiwan. A total amount of 3 kg sweet potato peel with a thickness of about 3 cm was collected and cut into pieces for freeze-drying, followed by grinding it into powder, placing into plastic bags for vacuum sealing and storing at −20 °C for use.

#### 2.1.1. Chemicals and Reagents

Carotenoid standards including all-trans-zeaxanthin and all-trans-β-cryptoxanthin were procured from Extrasynthese Co. (Genay, France), while all-trans forms of α-carotene, β-carotene, lutein and canthaxanthin (internal standard) were from Sigma-Aldrich (St. Louis, MO, USA).

The HPLC-grade solvents including methanol, ethanol, acetone, acetonitrile, methylene chloride and toluene were purchased from Lab-Scan Co. (Gliwice, Poland), while the analytical-grade n-hexane was from Grand Chemical Co. (Taipei, Taiwan). Deionized water was made using a Milli-Q water purification system from Millipore Co. (Bedford, MA, USA).

Tween 80 was obtained from Yu-Pa Co. (Taipei, Taiwan), while soybean oil was from a local supermarket (Taipei, Taiwan) and PEG 400 was from Sigma-Aldrich. Phosphotungstic acid (2%) was provided by the transmission electron microscope (TEM) lab located at the College of Medicine, Fu Jen Catholic University.

#### 2.1.2. Instrumentation

The HPLC-MS instrument (1100 series), composed of column temperature controller (G1316A), degasser (G1379A), quaternary pump (G1311A), binary pump (G1312A), photodiode array detector (G1315B) and quadrupole MS (6130) with multi-mode ion source (ESI and APCI), was from Agilent Technologies (Santa Clara, CA, USA). The on-line degassing system (DP-4010) was from Sanwatsusho Co. (Tokyo, Japan), while the injector (7161) was from Rheodyne Co. (California, CA, USA). The YMC carotenoid S-5 μm C30 column (250 × 4.6 mm ID, particle size 5 μm) was from Waters Co. (Milford, MA, USA). The transmission electron microscope (JEM-1400) was from JEOL Co. (Tokyo, Japan), while the dynamic light scattering (DLS) analyzer 90 plus was from Brookhaven Instruments Co. (Holtsville, NY, USA). The freeze-dryer (FD24) was from Chin-Min Co. (Taipei, Taiwan). The sonicator (2210R-DTH) was from Branson Co. (Danbury, CT, USA). The rotary evaporator (N-1200A) was from Eyela Co. (Tokyo, Japan), while the low-temperature circulation tank (Firstek B402L) was from Li-Chen Co. (Taoyuan, Taiwan).

#### 2.1.3. Cell Culture and Animal Study

Human breast cancer cell (MCF-7) and human breast epithelial cell (H184B5F5/M10) were obtained from Bioresource Collection and Research Center, Taiwan Food Industry Research and Development Institute (Hsinchu, Taiwan). Minimum Essential Medium (MEM) medium, α-MEM medium, fetal bovine serum (FBS), HEPES solution, sodium bicarbonate solution, sodium pyruvate, phosphate buffered saline (PBS), Hank’s balanced salt solution (HBSS) and 0.25% trypsin-EDTA were all from HyClone Co. (Logan, UT, USA).

Specific pathogen free (SPF) male BALB/c nude mice (four-weeks old) were purchased from the National Laboratory Animal Center (Taipei, Taiwan). The diet (LabDiet) was from LabDiet Co. (St. Louis, MO, USA). Both mouse EGF and VEGF ELISA kits were from Koma Biotech Co. (Seoul, Korea). Paclitaxel was from Sigma-Aldrich Co., while estradiol cypionate was from Da-Fong Drug Co. (Chanhua, Taiwan). The use of animals (mice) and the associated experimental procedures were approved by Fu Jen University Animal Care and Use Committee (Ethical permission approval number and date: A10704 and 23 April 2018).

### 2.2. Methods

#### 2.2.1. HPLC (High-Performance Liquid Chromatography) Analysis of Carotenoids in Sweet Potato Peel

A method based on Kao et al. [11] was modified to extract carotenoids from sweet potato peel. A 5-g sweet potato peel sample was mixed with 30 mL of hexane/ethanol/acetone/ toluene (10:6:7:7, *v/v/v/v*) in a flask, after which this mixture was shaken for 1 h and then 2 mL of 40% methanolic potassium hydroxide solution was added for saponification in the dark for 16 h. Next, 15-mL hexane was added and shaken for 10 min, followed by adding 15-mL of 10% sodium sulfate solution, shaking for one min and collecting the upper layer. Then 15-mL hexane was added to the lower layer and shaken for 10 min, and the upper layer was collected again. This procedure was repeated four times until the lower layer became colorless. All the supernatants from upper layers were combined and evaporated to dryness, followed by dissolving in 5 mL of methanol-methylene chloride (1:1, *v/v*), filtering through a 0.2-μm membrane filter and injecting 20 μL into HPLC.

A Waters YMC carotenoid S-5 μm C30 column (250 × 4.6 mm ID, 5 μm particle size) and a gradient mobile phase of (A) methanol/acetonitrile/water (74:14:12, *v/v/v*) and (B) methylene chloride (100%) based on Inbaraj et al. [12] was modified to separate various carotenoids in sweet potato peel: 70% A and 30% B initially, changed to 68% A in 5 min, 66% A in 10 min, 64% A in 15 min, 62% A in 18 min, 60% A in 20 min, 58% A in 24 min, 55% A in 27 min and maintained till 30 min. The flow rate was 1 mL/min and detection wavelength was 450 nm. Both retention factor (k) and separation factor (α) were used for evaluation of carotenoid separation efficiency. In addition, the absorption spectrum and purity of each peak was obtained automatically by a photodiode array detector.

The various carotenoids in sweet potato peel were identified by comparing retention times, MS spectra and the absorption spectra of unknown peaks with carotenoid standards. The MS spectra were determined using the HPLC-MS instrument (6130, Agilent Technologies, Santa Clara, CA, USA) with APCI mode with a scanning range of MW from 400–1200 *m*/*z*, drying gas flow 7 L/min, nebulizer pressure 10 psi, dry gas temperature 330 °C, vaporizer temperature 230 °C, capillary voltage 2000 V, charging voltage 2000 V, fragmentor voltage 200 V and corona current 4 μA. For further identification of cis isomers of carotenoids, carotenoid standards including all-trans forms of lutein, zeaxanthin, β-cryptoxanthin, β-carotene and α-carotene were each prepared with a concentration of 100 μg/mL and poured into a 10-mL transparent vial separately for photoisomerization [12]. All the vials were placed in an illumination chamber (25 °C) with four fluorescent tubes (55 cm long with 20 W each) hanging above, with the illumination distance 30 cm and an intensity of 2000–3000 lux. After 24-h illumination, all the carotenoid standard solutions were filtered through a 0.22-μm membrane filter and 20 μL of each standard was injected into HPLC. Then the retention time and absorption spectrum of each cis-isomer peak of carotenoids on the HPLC chromatogram was compared with those peaks of carotenoids in sweet potato peel. Also, the absorption spectra of cis isomers of carotenoids were used for identification based on several previous studies by Kao et al. [11] and Inbaraj et al. [12,13].

#### 2.2.2. Method Validation

The repeatability was performed by determining carotenoid concentration in sweet potato peel in the morning, afternoon and evening on the same day with three replicates each for a total of nine replicates, while the intermediate precision was carried out by analyzing carotenoid concentration in sweet potato peel in the morning, afternoon and evening on the first, second and third day for a total of nine analyses.

For recovery study, two concentrations (5 and 50 ppm) of mixed carotenoid standards were added to sweet potato peel sample for extraction and HPLC analysis, then the recovery data of each carotenoid was calculated based on the relative ratio of carotenoid amount after HPLC analysis to that before extraction and HPLC analysis.

Also, a series of carotenoid standard concentrations were prepared and injected into HPLC three times for determination of limit of detection (LOD) based on S/N ≥ 3, while the limit of quantitation (LOQ) was based on 3 × LOD.

#### 2.2.3. Quantitation of Carotenoids in Sweet Potato Peel

For quantitation, a fixed concentration (10 μg/mL) of internal standard all-trans-canthaxanthin was added to the sweet potato peel sample for extraction and HPLC analysis. Then the concentrations of individual carotenoid in sweet potato peel was calculated based on the linear regression equations of carotenoid standard curves, prepared by mixing various concentrations of carotenoid standards and a fixed concentration of internal standard and by plotting concentration ratio (carotenoid standard versus internal standard) against peak area ratio (carotenoid standard versus internal standard). The following formula was used for calculation of carotenoid contents in sweet potato peel:(1)$$\mathrm{Carotenoid}\;\mathrm{content}\;(\mu g/g)=\left(\frac{\mathrm{As}}{\mathrm{Ai}}\times a+b\right)\times \mathrm{Ci}\times V\times \mathrm{DF}\times \frac{1}{R}\times \frac{1}{\mathrm{Ws}}$$
where, As: peak area of carotenoid in sample; Ai: peak area of internal standard in sample; Ci: internal standard concentration; a: slope of the linear regression equation; b: intercept of the linear regression equation; V: final volume of extract; DF: dilution factor; R: recovery; Ws: sample weight.

#### 2.2.4. Preparation of Carotenoid Nanoemulsion

A carotenoid extract (7.268 g) containing carotenoid at 1375.84 μg/mL was poured into a tube and evaporated to dryness, followed by adding 0.1 g soybean oil (1%) and stirring, adding 0.6 g Tween 80 (6%), 0.2 g PEG 400 (2%) and 9.1% g deionized water (91%) for complete stirring and subsequent sonicating for 0.5 h. Next, a 10-mL nanoemulsion containing carotenoid at 1000 μg/mL was obtained.

#### 2.2.5. Determination of Carotenoid Nanoemulsion Characteristics

The average particle size and polydispersity index (PDI) of carotenoid nanoemulsion was determined by a dynamic light scattering analyzer (DLS) by collecting a portion of nanoemulsion (84 μL), diluting to 5 mL with 25 mM monopotassium phosphate solution, and filtering through a 0.22 μm membrane filter (Nylon, 13 mm) for determination.

The zeta potential was determined by collecting a portion of carotenoid nanoemulsion (100 μL) and diluting it 100-fold with deionized water, after which a sample (300 μL) was collected in a sample analysis tank for measurement of zeta potential at 25 °C with the potential range from −200 mV to 200 mV.

For TEM image determination, a sample of carotenoid nanoemulsion was collected and diluted 200-fold with deionized water, after which a 20-μL sample was collected and dropped onto a copper grid for sinking for 3 min, followed by removing the excessive sample with a glass filer paper, negative staining for 2 min with 20 mL phosphotungstic acid (2%), removing excessive phosphotungstic acid with a glass filter paper and placing it in a moisture-proof box for complete drying. Then the sample was enlarged 3 × 10^5^ times under 120 kVa for observation of particle size, shape and distribution.

The encapsulation efficiency of carotenoid in nanoemulsion was determined by collecting a portion of carotenoid nanoemulsion (100 μL), diluting one-fold with 25 mM monopotassium phosphate solution, pouring into a centrifuge tube containing 3 kDa dialysis membrane, centrifuging at 12,000 rpm for 20 min (25 °C), collecting the lower layer (200 μL) containing unencapsulated carotenoids, evaporating to dryness under nitrogen, dissolving in 100 mL methanol/methylene chloride (1:1, *v/v*) containing internal standard at 10 μg/mL and injecting into HPLC for quantitation. The encapsulation efficiency was then determined using the following formula:(2)$$\mathrm{Encapsulation}\;\mathrm{efficiency}\;\left(\%\right)=\frac{\mathrm{total}\;\mathrm{carotenoid}-\mathrm{unencapsulated}\;\mathrm{carotenoid}}{\mathrm{total}\;\mathrm{carotenoid}}\times 100$$

#### 2.2.6. Stability Test

The carotenoid nanoemulsion was stored at 25 °C for three months, during which a portion was collected every 15 days for determination of particle size and PDI by DLS and zeta potential by a zeta potential analyzer. Similarly, a sample of carotenoid nanoemulsion (200 μL) was collected in a tube and placed into a water bath for heating at 40 °C, 70 °C and 100 °C for 0.5, 1, 1.5 and 2 h. A total of 12 tubes were used for heating experiments and both particle size and zeta potential were determined.

#### 2.2.7. In Vitro Release Test

Gastric juice was prepared by mixing deionized water with 2 g/L sodium chloride and 3.2 g/L pepsin for pH control at 1.5 with the addition of hydrochloric acid. Then 8 mL gastric juice was mixed with 2 mL carotenoid nanoemulsion in a beaker, followed by stirring at 37 °C for 2 h (100 rpm), collecting one mL in a centrifuge tube, centrifuging at 1000 rpm (4 °C), collecting supernatant, evaporating to dryness and dissolving in 1 mL methanol-methylene chloride (1:1, *v/v*) for HPLC analysis.

Likewise, intestinal juice was prepared by mixing deionized water with 8.09% monopotassium phosphate and 5.16 g/L bile salts for pH control at 7.5 with the addition of sodium hydroxide. Next, 8 mL intestinal juice was mixed with 2 mL carotenoid nanoemulsion in a beaker, followed by stirring at 37 °C for 4 h (100 rpm), collecting one mL in a centrifuge tube, centrifuging at 1000 rpm (4 °C), collecting the supernatant, evaporating to dryness and dissolving in 1 mL methanol-methylene chloride (1:1, *v/v*) for HPLC analysis.

#### 2.2.8. Cell Culture Experiment

Human breast cancer cell MCF-7 was cultured in MEM medium containing 10% FBS while human breast epithelial cells H184B5F5/M10 were cultured in α-MEM medium containing 10% FBS. Cryogenic tubes were collected from liquid nitrogen and thawed at 37 °C, after which the cytosol containing 7% dimethyl sulfoxide (DMSO) was placed in a 10-cm plate, followed by adding 10 mL medium containing 10% FBS, incubating in a 37 °C incubator with 5% carbon dioxide and 100% relative humidity. After reaching 80% confluency, the medium was removed, followed by washing with PBS twice and adding 2 mL trypsin-EDTA (0.25%) for reaction in an incubator for a few minutes. After cell shedding, the same volume of medium was added to terminate the trypsin-EDTA reaction and transferred to a centrifuge tube for centrifuging at 1000 rpm for 5 min (25 °C). Then the supernatant was removed and one mL medium added for blowing cells evenly for subsequent cell seeding in a new plate by adding 8 mL fresh medium.

#### 2.2.9. MTT (3-[4,5-Dimethylthiazol-2-yl]-2,5 diphenyl tetrazolium bromide) Assay

Initially a 50-mg MTT powder was dissolved in 10 mL sterilized PBS solution and filtered through a 0.22-μm membrane filter to prepare a stock solution with a concentration of 5 mg/mL and stored at −20 °C. Prior to use, the MTT solution was diluted with HBSS at 1:9 (*v/v*). The MCF-7 and H184B5F5/M10 cells (1 × 10^4^ each) were then seeded separately in a 96-well plate. After 24 h attachment, the MEM or α-MEM medium was removed and washed with PBS, followed by adding five concentrations (2.5, 5, 10, 15 and 20 μg/mL) of carotenoid nanoemulsion or carotenoid extract for triplicate experiments. After 48 h incubation, the old medium was removed and washed with PBS. Then the MTT solution (200 μL) was added and reacted in the dark for 1 h, followed by adding DMSO (200 μL) to dissolve the purple crystal for absorbance measurement at 570 nm with an ELISA reader. The cell viability was calculated using a formula as described in a previous study [14]. The group without sample treatment was used as control and was incubated for 48 h.

#### 2.2.10. Cell Cycle Study

MCF-7 cells (1 × 10^6^) were seeded in a T75 flask and incubated overnight for cell attachment, after which the medium was removed and replaced with two concentrations (5 and 10 μg/mL) of carotenoid nanoemulsion or extract for triplicate experiments. After 48 h incubation, the medium was removed and washed with PBS, followed by adding 2 mL trypsin-EDTA for reaction in a 37 °C incubator for 3 min. After cell shedding, the reaction was terminated and cells collected in a centrifuge tube for centrifuging at 1200 rpm for 3 min (4 °C) and the supernatant was removed. Then PBS (300 μL) was added, followed by adding 702 μL of pre-cooled absolute alcohol slowly, storing overnight for cell fixation, collecting liquid for centrifuging at 1200 rpm for 5 min (4 °C), washing with PBS twice, and adding 800 μL of PBS containing 1% Triton-X 100 (Sigma-Aldrich), RNase (1 mg/mL) and propidium iodine (0.1 mg/mL) for reaction at 37 °C for 30 min. Then the proportions of sub-G1, G0/G1, S and G2/M phases were calculated using a Kaluza analysis software (version 3.1, Beckman Coulter Inc., Taipei, Taiwan).

#### 2.2.11. FITC-Annexin V/PI Study

MCF-7 cells (1 × 10^6^) were seeded in a T75 flask and incubated overnight for cell attachment, after which the medium was removed and replaced with two concentrations (5 and 10 μg/mL) of carotenoid nanoemulsion or extract for triplicate experiments. After 48 h incubation, the old medium was removed and washed with PBS. Then 2 mL trypsin-EDTA was added for reaction at 37 °C for 3 min until cell shedding. Following the reaction termination, cells were collected and centrifuged at 1200 rpm for 5 min (4 °C) for the supernatant removal, followed by washing with PBS once, centrifuging again at 1200 rpm for 5 min (4 °C), adding 0.1 mL of 1X binding buffer for cell suspension, 5 μL of FITC-Annexin V and 5 μL of propidium iodine for reaction at room temperature for 15 min. Then 0.4 mL of 1X binding buffer was added for analysis by a flow cytometry for calculating proportions of necrosis cells (Annexin V-/PI+), late-apoptosis cells (Annexin V+/PI+), viable cells (Annexin V-/PI-) and early-apoptosis cells (Annexin V+/PI-) by Kaluza analysis software.

#### 2.2.12. Determination of Caspase-3, Caspase-8 and Caspase-9

Activities of caspase-3, caspase-8 and caspase-9 were determined by a fluorometric assay-based kit as described in a previous study by Liu et al. [15].

#### 2.2.13. Animal Study

A total of 36 four-week-old SPF (specific pathogen free) female BALB/c nude mice were used in this study and raised in Fu Jen University Animal Research Center with one in an individual ventilated cage. The feeding temperature and relative humidity was 21 ± 2 °C and 55 ± 10%, respectively, with the light cycle 12 h. Both bedding material and sterilized water were replaced every week. All the experimental procedures followed the guideline approved by Fu Jen University Experimental Animal Care and Use Committee (Ethical permission approval number and date: A10704 and 23 April 2018).

Initially a dose of 15 mg/kg estradiol cypionate was injected into the neck skin of the mice, followed by injecting MCF-7 cells (5 × 10^6^) and matrigel (Merck KGaA, Darmstadt, Germany) (1:1) into the mice flank the next day for four weeks for tumor induction. The mice weight was recorded every week and tumor size measured by a caliper. The tumor volume was calculated using the following formula:(3)$$\mathrm{Tumor}\;\mathrm{volume}\;\left(V\right)=\frac{L\times W^{2}}{2}$$
where L: tumor length; W: tumor width.

For the animal experiment, a total of 36 mice were divided into six groups and both drugs and samples (carotenoid nanoemulsion and extract) were provided by intraperitoneal injection (IP) once every three days for a total of eight injections: (1) control–no sample or drug provided, (2) drug–injection of 0.2 mL paclitaxel each time, (3) carotenoid extract (low dose)–injection of 0.2 mL at 10 mg/kg body weight (BW) each time, (4) carotenoid extract (high dose)–injection of 0.2 mL at 20 mg/kg BW each time, (5) carotenoid nanoemulsion (low dose)–injection of 0.2 mL at 10 mg/kg BW each time, (6) carotenoid nanoemulsion (high dose)–injection of 0.2 mL at 20 mg/kg BW each time.

After completing the injections, all the mice were euthanized with carbon dioxide and tumors were collected for weight measurement. Also, blood was collected from the heart and then centrifuged at 3500 rpm for 20 min (4 °C) for serum collection. Both the amounts of epidermal growth factor (EGF) and vascular endothelial growth factor (VEGF) were determined using mouse EGF and VEGF kits, respectively. Briefly, a total of seven concentrations (15.625, 31.25, 62.5, 125, 250, 500 and 1000 pg/mL) of EGF or VEGF standards were each prepared and a portion (100 μL) was collected and added to a 96-well plate containing antibodies for reaction at room temperature for 2 h. Following washing with a wash buffer (300 μL) for four times, the solution (100 μL) containing 3,3′,5,5′-tetramethylbenzidine (TMB) substrate was added for 10 min reaction, followed by adding a stop solution (100 μL) to terminate the reaction and for the measuring of absorbance at 450 nm. Both the EGF and VEGF contents in the serum were then calculated based on the linear regression equations of the standard curves of EGF and VEGF prepared by plotting concentration against absorbance.

#### 2.2.14. Statistical Analysis

All of the data were subjected to statistical analysis by using a statistical analysis system [16]. Also, the analysis of variance was conducted by ANOVA for significance in mean comparison (*p* < 0.05) by Duncan’s multiple range test.

## 3. Results and Discussion

### 3.1. Analysis of Carotenoids in Sweet Potato Peel

Figure 1 shows the HPLC chromatogram of carotenoids extracted from sweet potato peel (TNG66) by using a C30 column and a gradient mobile phase of methanol-acetonitrile-water (74:14:12, *v/v/v*) and methylene chloride (100%) as described in the method section with a flow rate of 1 mL/min and a detection wavelength of 450 nm. A total of eleven carotenoids including all-trans-violaxanthin, cis-lutein, all-trans-lutein, all-trans-zeaxanthin, all-trans-canthaxanthin (internal standard), 15 or 15′-cis-β-cryptoxanthin, all-trans-β-cryptoxanthin, 15 or 15′-cis-β-carotene, 13 or 13′-cis-β-carotene, all-trans-β-carotene and all-trans-α-carotene were adequately separated within 29 min, with the retention factor ranging from 2.03–7.60, the separation factor ranging from 1.03–1.54 and the purity ranging from 90.7–99.7 (Table 1). The identification data including absorption spectra, Q-ratio and mass-to charge ratio (*m*/*z*) is shown in Appendix A [11,17,18,19]. The HPLC chromatograms of all-trans-carotenoid standards including lutein, zeaxanthin, β-cryptoxanthin, β-carotene and α-carotene during illumination at 25 °C for 24 h are shown in Appendix A. By comparison of retention time, absorption spectra and Q ratios of all-trans plus cis forms of various carotenoid standards after illumination with those peaks of sweet potato peel samples in the HPLC chromatogram, the various cis isomers of carotenoids were also identified (Appendix A [11,12,17,18,20,21]. Compared to a previous study by Kao et al. [11] employing a C30 column and a gradient mobile phase of methanol-acetonitrile-water and methylene chloride (100%) to separate 25 carotenoids in *Taraxacum formosanum* within 66 min, the retention time was much shorter by using a modified mobile phase in our study. Furthermore, both the retention factor and the separation factor shown in Table 1 revealed that an appropriate solvent strength of the mobile phase and a suitable selectivity of the mobile phase to carotenoid components in sweet potato peel sample was attained. In addition, the peak purities of all the carotenoid peaks were >90%.

For method validation and quantitation, the standard curves of all-trans forms of lutein, zeaxanthin, β-cryptoxanthin, β-carotene and α-carotene were prepared with the linear regression equations being y = 0.8493x − 0.0538, y = 0.8295x − 0.065, y = 3.5643x − 0.1506, y = 4.2761x + 0.3222 and y = 2.2856x − 0.2224, respectively, and the coefficient of determination (R^2^) all being higher than 0.99. The LOD of all-trans forms of lutein, zeaxanthin, β-cryptoxanthin, β-carotene and α-carotene was 0.1, 0.1, 0.025, 0.05 and 0.05 μg/mL, respectively, while the LOQ was 0.3, 0.3, 0.075, 0.15 and 0.15 μg/mL.

Table 2 shows the quality control data of carotenoids in sweet potato peel, with the relative standard deviation (RSD) of repeatability being from 0.3–2.1% and the intermediate precision being from 0.3–4.2%. The recovery data of all-trans forms of lutein, zeaxanthin, β-cryptoxanthin, β-carotene and α-carotene was 94.6, 92.1, 91.9, 94.7 and 86.1%, respectively (Table 2). All the method validation data met the guideline requirement set by Taiwan Food and Drug Administration (TFDA) [22], implying that a reliable method for determination of carotenoids in sweet potato peel was established. Following quantitation, all trans-β-carotene was present in the highest amount (663.8 μg/g) in sweet potato peel, followed by 13- or 13′- β-carotene (255.8 μg/g), all-trans-β-cryptoxanthin (155.9 μg/g), all-trans-α-carotene (75.59 μg/g), 15 or 15′-cis-β-cryptoxanthin (72.37 μg/g), 15 or 15′-cis-β-carotene (65.67 μg/g), all-trans-lutein (42.93 μg/g), all-trans-zeaxanthin (25.57 μg/g), cis-lutein (18.35 μg/g) and all-trans-violaxanthin (5.48 μg/g) (Table 1).

In several previous studies, Chen [1] determined carotenoids in three sweet potato cultivars and reported that the carotenoid contents in sweet potato pulp and peel were similar, with the major carotenoid being β-carotene (271.80–450.88 μg/g, DW). However, the individual carotenoid remained undetermined. Similarly, the carotenoid contents in seven cultivars of sweet potato were reported to be from 1.02–61.94 μg/g [23], while the total carotenoid content was 24.31 μg/g FW in sweet potato tubers [24]. By comparison, the total carotenoid content in sweet potato peel was much higher, which was probably caused by the difference in the sweet potato cultivar and the analytical method used in our study.

### 3.2. Preparation of Carotenoid Nanoemulsion

In this study, PEG 400, Tween 80 and soybean oil were used as the major components to prepare carotenoid nanoemulsion with carotenoid extract as raw material from sweet potato peel. PEG 400 was selected as a surfactant based on a previous study by Jing et al. [25], reporting that the elimination of drug from reticuloendothelial system was minimized during drug delivery, while a lower toxicity was observed for paclitaxel nanoemulsion prepared from PEG 400. Also, Tween 80 was shown to be an effective surfactant by lowering surface tension and elevating kinetic stability in the gastrointestinal tract [26], while highly unsaturated soybean oil was reported to enhance carotenoid stability and protect the carotenoid from degradation [27,28]. Thus, a transparent carotenoid nanoemulsion was successfully prepared by using an appropriate ratio of soybean oil (1%), Tween 80 (6%), PEG 400 (2%) and deionized water (91%) (Figure 2). The mean particle size and polydispersity index (PDI) as determined by DLS was 13.3 nm and 0.238, respectively, while the mean particle size determined by TEM was 15.7 nm (Figure 2). This outcome indicated that a narrow and even distribution of nanoparticles was obtained due to a low PDI of 0.238 [29]. A high negative zeta potential (−69.8 mV) also implied that a high stability of carotenoid nanoemulsion was prepared, as it was reported that the zeta potential of the solution must be >30 mV or <−30 mV to maintain a high stability [30]. Compared to the β-carotene nanoemulsion prepared by Qian et al. [31] and Jo and Kwon [32], a much lower zeta potential was obtained in our study and thus a higher stability was attained. In addition, the encapsulation efficiency was calculated to be 97%, which should also contribute to the high stability of the carotenoid nanoemulsion prepared from sweet potato peel.

### 3.3. Stability of Carotenoid Nanoemulsion

Table 3 shows the mean particle size, PDI and zeta potential of carotenoid nanoemulsion over a 90-day storage period at 25 °C. Only a minor change in particle size, PDI and zeta potential was observed during storage for 90 days at 25 ° C. Likewise, during heating at 40, 70 and 100 °C for 2 h, the mean particle size only showed a slight change, while the zeta potential declined following a rise in heating temperature and time length (Table 3). Nevertheless, the zeta potential was still <−30 mV, indicating that the carotenoid nanoemulsion stability was maintained even after the heating temperature reached 100 °C for 2 h. This finding further demonstrated that the carotenoid nanoemulsion prepared from sweet potato peel in this study possessed both storage and heating stability.

### 3.4. In Vitro Release Study

Figure 3 shows the in vitro release of carotenoid nanoemulsion as a function of incubation time in simulated gastric fluid (SGF) for 4 h and simulated intestinal fluid (SIF) for 10 h. Following the treatment of carotenoid nanoemulsion with SGF for 1.5 h, the in vitro release reached 18.6% and became flattened thereafter with a maximum release of 22.5% at 4 h. However, following the treatment of the carotenoid nanoemulsion with SIF for 4 h, the in vitro release reached 43.4% and became flattened afterwards, followed by a maximum release of 54.7% at 10 h. This finding demonstrated that the carotenoid nanoemulsion prepared in this study was mainly released in the intestine. In a similar study, Tan et al. [33] prepared carotenoid liposome and reported that the in vitro release of β-carotene was 21% and 50% following treatment with SGF for 4 h and SIF for 6 h, respectively. Likewise, Yi et al. [34] prepared β-carotene nanoparticle from β-lactoglobulin-dextran and the in vitro β-carotene release was shown to be 6.2% and 5.4%, respectively, following treatment with SGF for 2 h, as well as 51.8% and 60.9%, respectively, with SIF for 2 h. Apparently, the smaller the particle size, the longer the release time. It is possible that following the release of carotenoid from nanoemulsion in the intestine, the carotenoid could form micelles (4–60 nm) at the duodenum for absorption by intestinal epithelial cells to enhance carotenoid bioavailability.

### 3.5. Cell Culture Study

To avoid the interference of solvent and nanoemulsion composition on growth of MCF-7 and H184B5F5/M10 cells, the effect of blank nanoemulsion without carotenoids (A) and DMSO (B) on growth of human breast cancer cells MCF-7 and mammary cells H184B5F5/M10 was studied (Figure 4). Following treatment of blank nanoemulsion at a dose of 0.05, 0.075 and 0.1%, the MCF-7 cell viability was 118.4, 114.5 and 108.2%, respectively. However, the MCF-7 cell viability significantly declined (*p* < 0.05) to 94.4, 82.5 and 76%, respectively, after the blank nanoemulsion dose was raised to 0.125, 0.25 and 0.5%. Similarly, the H184B5F5/M10 cell viability was 101.9, 101.3, 100.5 and 98.9%, respectively, following the treatment of the blank nanoemulsion at a dose of 0.05, 0.075, 0.1 and 0.125%, but significantly decreased (*p* < 0.05) to 96.5 and 95.7%, respectively, for a higher dose at 0.25 and 0.5%. Apparently, a dose-dependent decline in cell viability of MCF-7 and H184B5F5/M10 was shown for the blank nanoemulsion treatments. A similar outcome was observed for the DMSO treatment, with the MCF-7 cell viability being 106, 102.1 and 100.8%, respectively, after the DMSO treatment at a dose of 0.1, 0.2, and 0.5%, but significantly decreased (*p* < 0.05) to 95.2, 90.4 and 85.6%, respectively, when the dose was raised to 1, 1.5 and 2%. Likewise, the H184B5F5/M10 cell viability was 102.2, 100.3, 97.3 and 98.9%, respectively at a DMSO dose of 0.1, 0.2, 0.5 and 1%, but significantly declined (*p* < 0.05) to 95.4 and 93.2% for a higher dose at 1.5 and 2%, respectively. Like blank nanoemulsion, a dose-dependent decrease in viability of MCF-7 and H184B5F5/M10 cells was also shown for the DMSO treatment. Thus, for subsequent experiments, both blank nanoemulsion and DMSO doses were controlled at 0.25 and 2%, respectively, to maintain the high viability of both the MCF-7 and H184B5F5/M10 cells.

### 3.6. Growth of MCF-7 and H184B5F5/M10 Cells as Affected by Carotenoid Nanoemulsion and Extract

Figure 5 shows the effect of carotenoid nanoemulsion and extract on growth of MCF-7 cells (A and B) and H184B5F5/M10 cells (C and D). A dose-dependent decline in viability of MCF-7 and H184B5F5/M10 cells was shown for both carotenoid nanoemulsion and extract treatments. More specifically, the MCF-7 cell viability significantly dropped (*p* < 0.05) to 78.8, 66.5, 47.3, 36.9 and 29.7%, respectively, when treated with carotenoid nanoemulsion at a dose of 2.5, 5, 10, 15, and 20 μg/mL, as well as 85.1, 70.4, 51.6, 46.6 and 40.4% when treated with carotenoid extract at the same doses. Comparatively, carotenoid nanoemulsion was more effective in inhibiting the growth of MCF-7 cells than carotenoid extract, as evident by an IC_50_ of 9.8 and 14.2 μg/mL, respectively. Similarly, following the treatment of carotenoid nanoemulsion at a dose of 2.5, 5, 10, 15 and 20 μg/mL, the H184B5F5/M10 cell viability significantly decreased (*p* < 0.05) to 90.6, 88.4, 74.2, 65.4 and 56.2%, respectively, as well as to 99.9, 94.2, 83.5, 76.2 and 65.4% when treated with carotenoid extract at the same doses. By comparison, carotenoid nanoemulsion possessed a higher toxicity towards H184B5F5/M10 cells than carotenoid extract. In several previous studies, Cui et al. [35] studied the effect of β-carotene on growth of MCF-7 cells and a time- and dose-dependent decrease was shown with an IC_50_ of 5 μM. In a later study Shree et al. [36] extracted and purified β-carotene from spinach and studied its effect on the growth of MCF-7 cells; a cell viability of 64% was observed when treated with β-carotene at a dose of 5 μM. However, no significant difference (*p* > 0.05) in human fibroblast cell (HDFa) viability was shown when treated with β-carotene at 0.5–50 μM. However, in another study dealing with the effect of β-carotene (10 μM) on viability of breast cancer cells MCF-7, MDA-MB-235 and MDA-MB-231, the cell viability was 60, 70 and 30%, respectively, after 48 h incubation. All the outcomes revealed that the inhibition efficiency of breast cancer cells can be dependent upon carotenoid variety and dose, cell type, and incubation condition. By comparison, both carotenoid nanoemulsion and extract prepared in this study showed a higher inhibition effect than β-carotene reported in the literature, which was probably caused by the synergistic effect of various carotenoids present in the nanoemulsion and extract.

### 3.7. Cell Cycle Study

The cell cycle distribution of MCF-7 as affected by carotenoid extract and nanoemulsion is shown in Table 4. Compared to control, the proportions of sub-G1 and G0/G1 phases significantly increased (*p* < 0.05) by 2.09 and 2.56%, respectively, while that of S and G2/M phases declined by 2.53 and 2.02%, respectively, following treatment with carotenoid extract at 5 μg/mL. A similar outcome was observed for carotenoid extract at 10 μg/mL, with the sub-G1 and G0/G1 phases being significantly raised (*p* < 0.05) by 2.41 and 4.63%, respectively, but decreased by 2.43 and 4.03% for S and G2/M phases, respectively. Carotenoid nanoemulsion showed a similar trend with sub-G1 and G0/G1 phases increased by 2.8 and 6.31%, respectively, at a dose of 5 μg/mL, compared to control, while S and G2/M phases diminished by 3.05 and 6.21%, respectively. However, at a high dose of 10 μg/mL, sub-G1 and G0/G1 phases rose by 2.41 and 7.53%, respectively, compared to control, but dropped by 3.6 and 6.41% for S and G2/M phases, respectively. By comparison, at the same dose, the carotenoid nanoemulsion could result in a higher proportion of sub-G1 and G0/G1 phases, but a lower proportion of S and G2/M phases than carotenoid extract. This finding suggested that both carotenoid nanoemulsion and extract could lead to the apoptosis of MCF-7 cells with a possible arrest at the G0/G1 phase. In several previous studies, Gloria et al. [37] also reported that following treatment of breast cancer cells with β-carotene (5 and 10 μM), both MCF-7 and MDA-MB-235 cells were arrested at G0/G1, while MDA-MB-231 cells arrested at G2/M. A similar outcome was reported for the arrest of MCF-7 cells at the G0/G1 phase when treated with lycopene [38] and β-carotene extracted from spinach [39]. Obviously the arrest of breast cancer cells at a certain phase can be dependent upon carotenoid variety, cell type, incubation condition and dose. In our study we demonstrated that the carotenoid nanoemulsion was more effective than the carotenoid extract in arresting MCF-7 cell cycle at the G0/G1 phase.

### 3.8. Analysis of Cell Apoptosis

Figure 6 shows the apoptosis of MCF-7 cells as affected by carotenoid nanoemulsion and extract. Compared to control, the proportions of necrosis, late apoptosis and early apoptosis cells rose by 0.68, 9.39 and 8.11%, respectively, following carotenoid extract treatment at 5 μg/mL, while that of viable cells dropped by 18.16%. Likewise, at a high dose of 10 μg/mL, the proportions of necrosis, late apoptosis and early apoptosis cells further increased by 0.88, 12.37 and 10.43%, respectively, while that of viable cells diminished by 23.68%. Carotenoid nanoemulsion showed a similar trend with the proportions of necrosis, late apoptosis and early apoptosis cells being raised by 0.84, 7.97 and 5.6%, respectively, at a dose of 5 μg/mL when compared to control, while that of viable cells decreased by 14.4%. Similarly, at a high dose of 10 μg/mL, the proportions of necrosis, late apoptosis and early apoptosis cells further climbed by 1.45, 20.1 and 17.95%, respectively, compared to control, while that of viable cells declined by 39.49%. All of the results revealed that both carotenoid nanoemulsion and extract could lead to apoptosis of MCF-7 cells through the elevation of proportions of necrosis, late apoptosis and early apoptosis cells, with the former being more efficient than the latter. Comparatively, the late apoptosis cells were raised to a higher proportion than the necrosis and early apoptosis cells following the treatment of carotenoid nanoemulsion and extract at 5 or 10 μg/mL. Also, a dose-dependent increase in proportions of necrosis, late apoptosis and early apoptosis cells was shown for both carotenoid nanoemulsion and extract treatments. Similar findings were reported for MCF-7 cells when treated with β-carotene and lycopene [37] as well as β-carotene extracted from spinach [39].

### 3.9. Activities of Caspase-3, Caspase-8 and Caspase-9

The effect of carotenoid extract and nanoemulsion on caspase-3, caspase-8 and caspase-9 activities of MCF-7 cells is shown in Figure 7. Compared to control, the caspase-3 activity significantly rose (*p* < 0.05) by 1.4-, 1.46- and 1.84-fold, respectively, following treatment of carotenoid extract at 10 μg/mL and nanoemulsion at 5 and 10 μg/mL. A similar trend was observed for caspase-8 and caspase-9 activities, with the former being significantly raised (*p* < 0.05) by 1.1-, 1.42-, and 2.1- and 2.4-fold for carotenoid extract at 5 and 10 μg/mL and nanoemulsion at the same doses, respectively, compared to control, while the latter increased by 1.22-, 1.46-, 1.64- and 2.27-fold for the same treatment, respectively. Apparently, a dose-dependent rise was shown for all the activities of caspase-3, caspase-8 and caspase-9. By comparison, at the same dose, the carotenoid nanoemulsion resulted in higher activities of caspase-3, caspase-8 and caspase-9 than carotenoid extract. It has been well established that caspase can be activated to promote apoptosis through the mitochondria, the death receptor or the endoplasmic reticulum pathway, with caspase-3 being responsible for apoptosis execution and both caspase-8 and caspase-9 for apoptosis initiation. Additionally, caspase-8 can be activated through an external route (death receptor), while caspase-9 is activated through an internal route (mitochondria). Thus, in our study we demonstrated that both carotenoid extract and nanoemulsion could elevate caspase-8 and caspase-9 activities for caspase-3 activation leading to apoptosis of MCF-7 cells, with the carotenoid nanoemulsion being more efficient than the carotenoid extract. A similar outcome was reported for MCF-7 cells when treated with β-carotene extracted from spinach [39].

### 3.10. Animal Study

No significant difference (*p* > 0.05) in mice body weight was shown following the four-week treatment of carotenoid extract or nanoemulsion compared to control. However, the paclitaxel treatment resulted in a significant decline (*p* < 0.05) in mice body weight from 21.2 g (control) to 20.3 g, implying a possible side effect of paclitaxel. Figure 8 shows the effect of carotenoid extract, nanoemulsion and paclitaxel on tumor volume, size and weight of nude mice. Compared to control, paclitaxel was the most effective in reducing tumor volume by 75.4%, followed by carotenoid nanoemulsion at 10 μg/mL (65%) and 5 μg/mL (49.7%), carotenoid extract at 10 μg/mL (46.7%) and 5 μg/mL (26.5%). Apparently, a dose-dependent decrease in tumor volume was shown for both carotenoid nanoemulsion and extract treatments, with the former being more efficient than the latter. The same outcome was observed for tumor size and weight. Compared to control, the tumor weight was reduced by 77.4, 56.2, 40.3, 36.1 and 18.7%, respectively, following the treatment of paclitaxel, carotenoid nanoemulsion (10 and 5 μg/mL) and extract (10 and 5 μg/mL), with the carotenoid nanoemulsion being more effective than the extract. In a previous study dealing with the effect of palm oil carotene on breast cancer tumorigenicity in nude mice, Nesaretnam et al. [40] reported that the MCF-7 cells-induced tumor volume was reduced by 50% while the tumor incidence rate dropped by 40%. Similarly, lutein prepared from marigold extract was reported to reduce WAZ-2T- (breast cancer cell) induced tumor volume and weight by 67.4 and 86.6%, respectively [41]. Furthermore, the tumor incidence rate was decreased by 30%. In another study dealing with the effect of the combination of lycopene and genistein on 7,12-dimethyl benz(a)anthracene-induced breast cancer in rats, tumor weight and volume was reduced by 48 and 18%, respectively, following 20-week administration [42]. Likewise, the tumor volume in rats was reduced by 41 and 79.41%, respectively, following 16-week oral administration of lyophilized carotenoid-rich marine alga at 500 and 1000 mg/kg BW [43]. Apparently the inhibition efficiency of tumors in mice or rats can be dependent upon carotenoid variety, preparation method of carotenoid, dose, administration length and cancer cell type.

### 3.11. Serum Growth Factor in Nude Mice

Both epidermal growth factor (EGF) and vascular endothelial growth factor (VEGF) play a vital role in tumor formation, with the former being capable of inhibiting cancer cell apoptosis through the conjugation of EGF and its receptor for activation of protein expression controlling apoptosis, and the latter being able to promote angiogenesis through the activation of vascular endothelial cells for formation of new blood capillaries and the provision of nutrients for tumor growth. Figure 9 shows the effect of carotenoid extract, nanoemulsion and paclitaxel on EGF and VEGF levels in nude mice serum. A dose-dependent decline in EGF and VEGF levels was shown for both carotenoid extract and nanoemulsion treatments, with the latter being more pronounced than the former. Comparatively, paclitaxel was the most effective in decreasing both EGF and VEGF levels in mice serum, followed by carotenoid nanoemulsion at 20 and 10 μg/mL, as well as carotenoid extract at 20 and 10 μg/mL. This finding clearly demonstrated that both carotenoid extract and nanoemulsion prepared from sweet potato peel were effective in reducing tumor formation induced by MCF-7 breast cancer cells through a decline of both EGF and VEGF levels in mice serum. In a previous study, Kubatka et al. [44] studied the antineoplastic effects of carotenoids-rich *Chlorella pyrenoidosa* on breast tumors in rats induced by *N*-methyl-*N*-nitrosourea, and the EGFR-2 level was reduced by 19% following 14-weeks of administration, implying that the incorporation of *Chlorella pyrenoidosa* into feed was effective in retarding angiogenesis. Similarly, the expression of VEGF-α mRNA was significantly reduced through the up-regulation of the Wnt/β-catenin pathway following the treatment of 4T1 breast cancer cells with carotenoids prepared from saffron [45].

Clinically, breast cancer is categorized into three major subtypes based on the presence or absence of estrogen receptors (ER), progesterone receptors (PR), and human epidermal growth factor receptor-2 (HER2), with more than 60% of breast cancers being ER-positive and about 20% being negative for ER, PR and HER, also named as triple-negative breast cancer (TNBC) [46,47]. As TNBC cells such as BT20, MDA468, MDA231 and MDA436 lack specific cell-surface receptors for active targeting, the treatment of TNBC remains incurable using currently available drugs. In a study dealing with metabolic alterations in TNBC cells in comparison with other subtypes of breast cancer cells using molecular and metabolic analyses, Pelicano et al. [46] demonstrated that, compared to MCF-7 cells (ER/PR double positive and HER-2 negative), TNBC cells possessed special metabolic characteristics manifested by high glucose intake, increased lactate production and low mitochondrial respiration, suggesting that this metabolic intervention may be an effective therapeutic strategy for TNBC cells. Nevertheless, there are some other approaches which may be effective in curing TNBC. For instance, the development of novel drug-nanoparticles capable of accumulating in patient-derived xenograft tumors through the enhanced permeability and retention (EPR) effect in tumor vasculature may lead to new and effective TNBC treatments [48]. In other words, the preparation of carotenoid nanoemulsion from sweet potato peel in our study may be used to treat ER-, PR- or HER-2-positive tumors and TNBC through the passive targeting effect.

Furthermore, it was reported that the endothelial cells in the blood vessels of normal tissues may have gaps which are approximately 7–10 nm in size, but these gaps may increase to a few hundred nanometers in the tumor micro-vasculature to permit nanoparticle accumulation through the EPR effect [49]. Therefore, the carotenoid nanoemulsion with a size of 15.7 nm based on TEM analysis in this study should be capable of penetrating from the extracellular matrix into the cytoplasm and nucleus for antitumor efficiency. As pointed out by Farokhzad and Langer [50] and Huang et al. [49], the tumor tissue accumulation is a passive process requiring long circulation to facilitate the time-dependent extravasations of drug delivery systems through the leaky tumor microvasculature for accumulation. In addition, the elimination of nanoparticles in carotenoid nanoemulsion from the reticuloendothelial system was avoided with nanoparticle size <200 nm [51]. However, in this study we only measured tumor size and volume as well as EGF and VEGF in mice serum. Thus, for future studies, it is important to examine tumor tissue to find out if the same effect of apoptosis and cell cycle is shown.

## 4. Conclusions

In conclusion, the carotenoid nanoemulsion prepared from sweet potato peel was effective in inhibiting MCF-7 breast cancer cells and reducing breast tumor volume and weight, possibly through a passive targeting effect. A high heating and storage stability was shown for the carotenoid nanoemulsion composed of soybean oil, Tween 80, PEG 400 and deionized water. HPLC analysis revealed the presence of 10 carotenoids in sweet potato peel, with β-carotene present in the highest amount. The anti-tumor efficiency may be due to the synergistic effect of β-carotene and the other carotenoids in nanoemulsion.

## Figures and Tables

**Figure 1 pharmaceutics-14-00980-f001:**
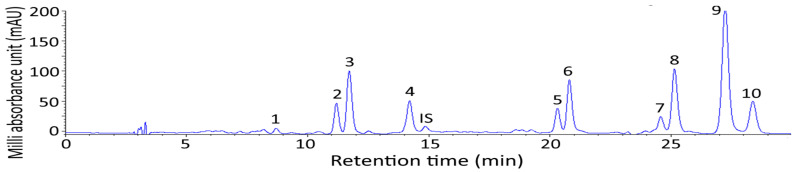
HPLC chromatogram of carotenoids extracted from sweet potato peel (TNG66). Column, C30; gradient mobile phase of methanol-acetonitrile-water (74:14:12, *v/v/v*), and dichloromethane (100%); flow rate, 1 mL/min; detection wavelength, 450 nm. Peaks: 1–All-trans-violaxanthin, 2–cis-lutein, 3–All-trans-lutein, 4–All-trans-zeaxanthin, IS–All-trans-canthaxanthin, 5–15- or 15′-cis-β-cryptoxanthin, 6–All-trans-β-cryptoxanthin, 7–15- or 15′-cis-β-carotene, 8–13- or 13′-cis-β-carotene, 9–All-trans-β-carotene, 10–All-trans-α-carotene.

**Figure 2 pharmaceutics-14-00980-f002:**
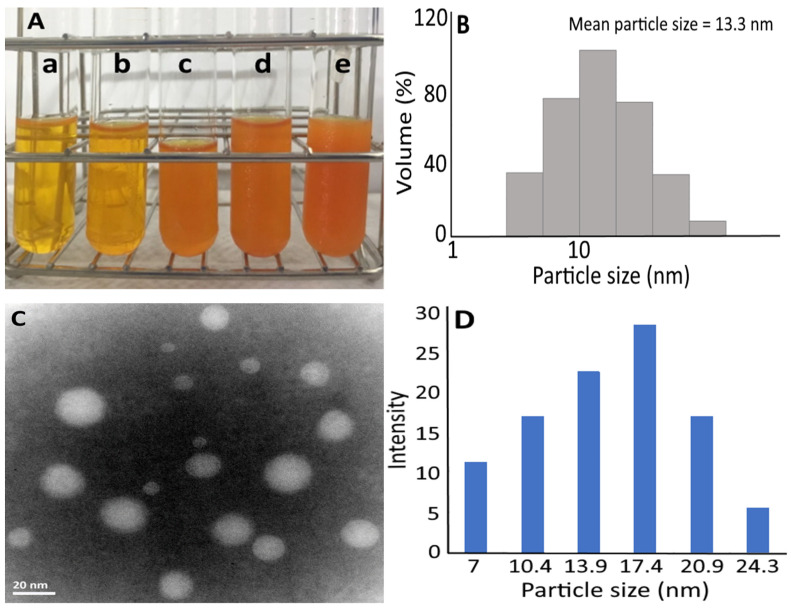
Appearance of carotenoid nanoemulsions prepared at different concentrations: a—1000 ppm, b—3000 ppm, c—5000 ppm, d—8000 ppm, e—10,000 ppm (**A**) along with particle size distribution as determined by a dynamic light scattering method (**B**) as well as their TEM image (**C**) and associated particle size distribution (**D**) from 10,000 ppm of carotenoid nanoemulsion.

**Figure 3 pharmaceutics-14-00980-f003:**
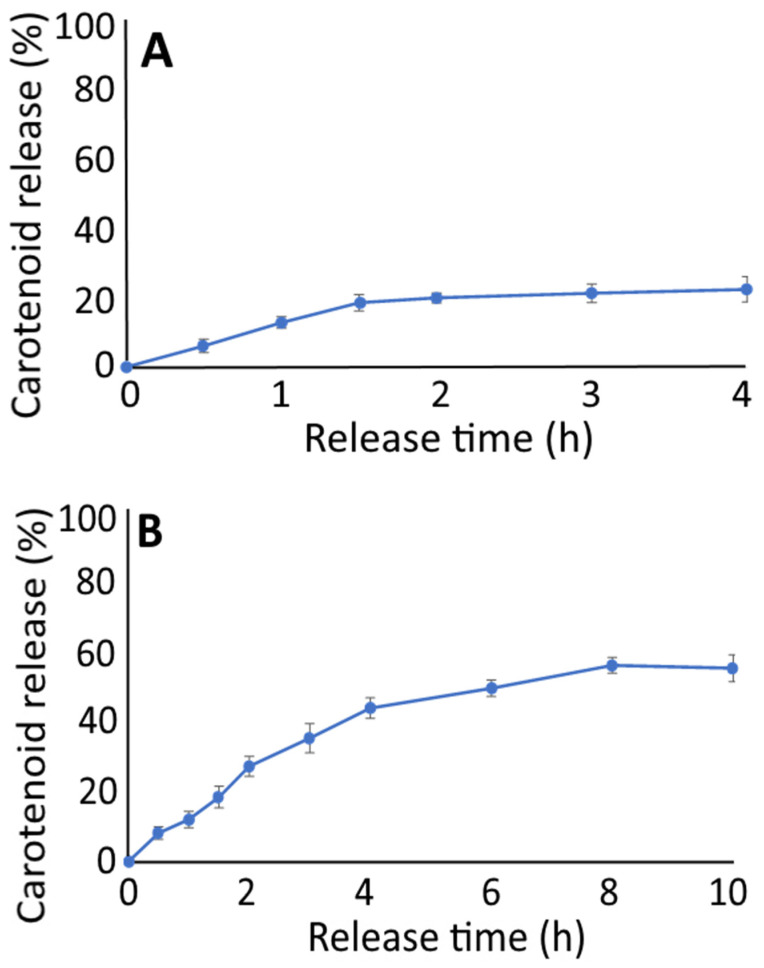
In vitro carotenoid release from carotenoid nanoemulsions as a function of incubation time in simulated gastric fluid for 4 h (**A**) and simulated intestinal fluid for 10 h (**B**). Results are presented as mean ± standard deviation of triplicate measurements.

**Figure 4 pharmaceutics-14-00980-f004:**
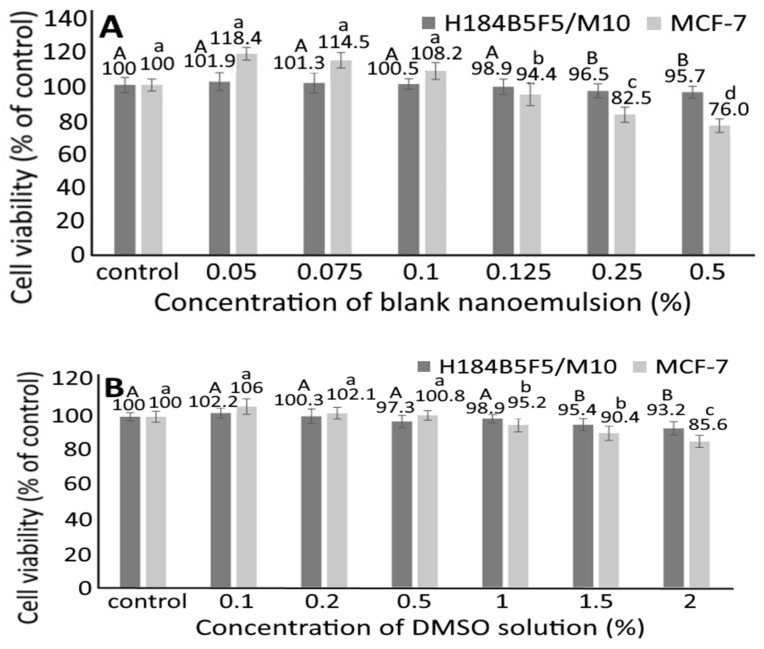
Effect of blank nanoemulsion (**A**) and DMSO solution (**B**) on growth of human breast MCF-7 cancer cells and mammary H184B5F5/M10 cells. Results are presented as mean ± standard deviation of triplicate independent experiments. Data with different capital letters (A,B) for H184B5F5/M10 cells and small letters (a–d) for MCF-7 cancer cells are significantly different at *p* < 0.05.

**Figure 5 pharmaceutics-14-00980-f005:**
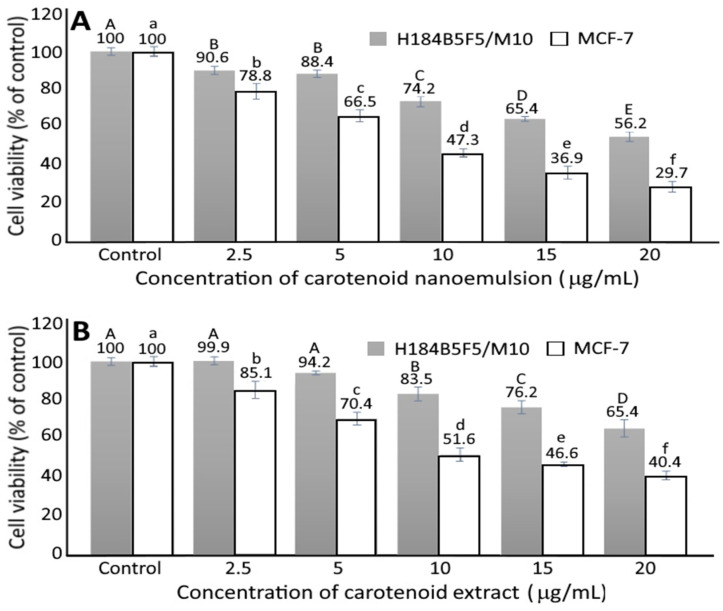
Effect of carotenoid nanoemulsions (**A**) and carotenoid extracts (**B**) on growth of human breast MCF-7 cancer cells and human mammary H184B5F5/M10 cells. Results are presented as mean ± standard deviation of triplicate independent experiments. Data with different capital letters (A–E) for H184B5F5/M10 cells and small letters (a–f) for MCF-7 cells are significantly different at *p* < 0.05.

**Figure 6 pharmaceutics-14-00980-f006:**
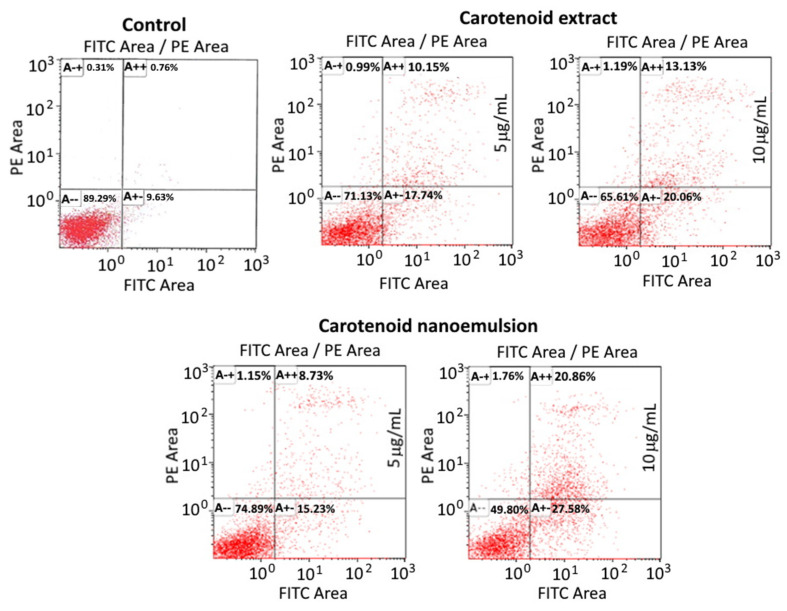
Apoptosis of MCF-7 cancer cell line as affected by carotenoid extracts and carotenoid nanoemulsions. Control, cells were incubated in medium only. A−+, necrosis cells; A++, late apoptosis cells, A−−, viable cells; A+−, early apoptosis cells.

**Figure 7 pharmaceutics-14-00980-f007:**
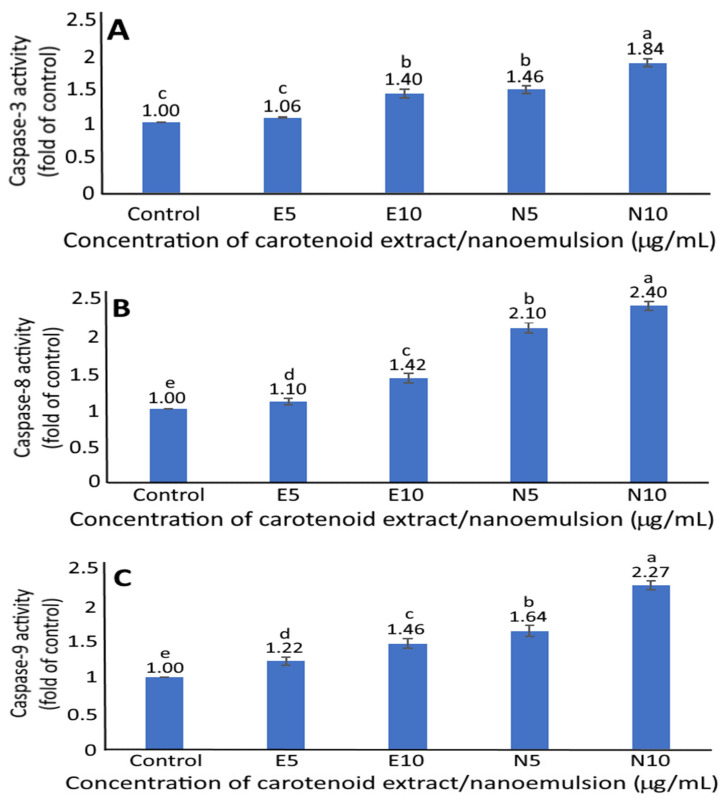
Effect of carotenoid extracts and nanoemulsions on caspase-3 (**A**), caspase-8 (**B**) and caspase-9 (**C**) activities of human breast cancer cell MCF-7. Control cells were incubated in medium only; E5 and E10, carotenoid extract concentration respectively at 5 and 10 μg/mL; N5 and N10, carotenoid nanoemulsion concentration respectively at 5 and 10 μg/mL. Results are presented as mean ± standard deviation of triplicate independent experiments. Data with different small letters (a–e) are significantly different at *p* < 0.05.

**Figure 8 pharmaceutics-14-00980-f008:**
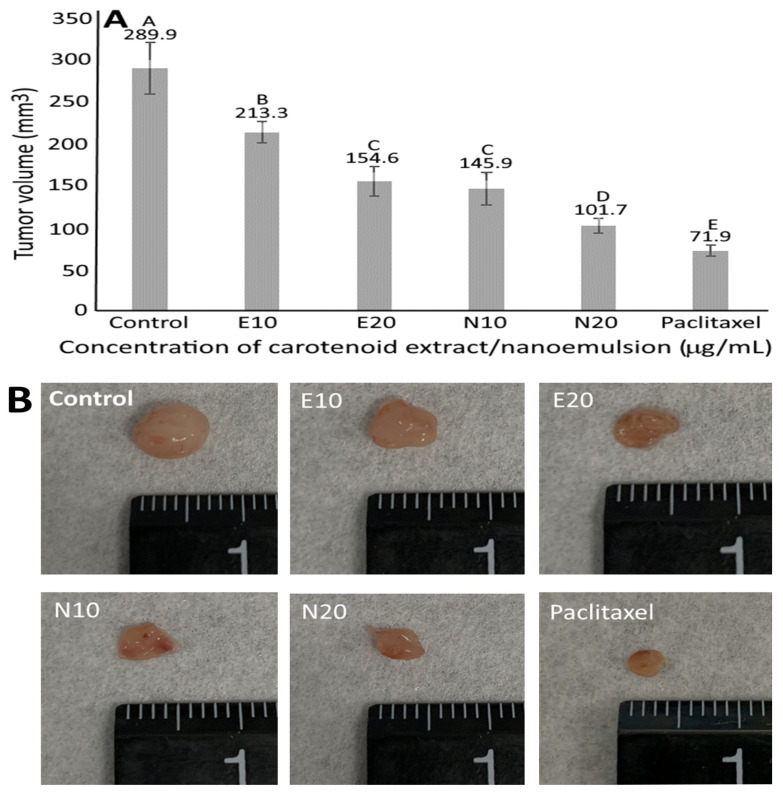
Effect of carotenoid extracts, carotenoid nanoemulsions and paclitaxel on tumor volume (**A**), tumor size (**B**) and tumor weight (**C**) of nude mice. E10 and E20, carotenoid extract concentration respectively at 10 and 20 μg/mL; N10 and N20, carotenoid nanoemulsion concentration respectively at 10 and 20 μg/mL; Paclitaxel, paclitaxel concentration at 10 μg/mL. Results are presented as mean ± standard deviation of measurements from six nude mice (*n* = 6). Data with different capital letters (A–F) are significantly different at *p* < 0.05.

**Figure 9 pharmaceutics-14-00980-f009:**
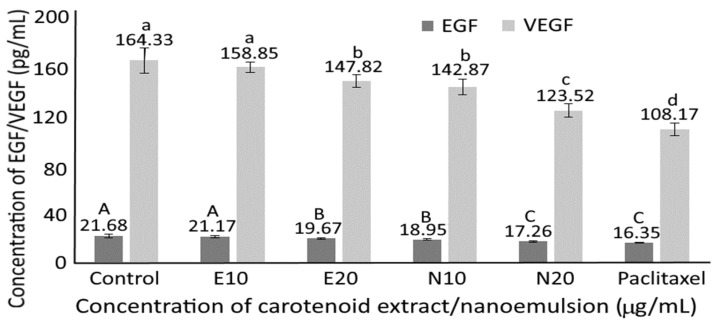
Effect of carotenoid extracts, carotenoid nanoemulsions and paclitaxel on serum EGF and VEGF levels in nude mice. E10 and E20, carotenoid extract concentration respectively at 10 and 20 μg/mL; N10 and N20, carotenoid nanoemulsion concentration respectively at 10 and 20 μg/mL; Paclitaxel, paclitaxel concentration at 10 μg/mL. Results are presented as mean ± standard deviation of measurements from six nude mice (*n* = 6). Data with different capital letters (A–C) for EGF and small letters (a–d) for VEGF are significantly different at *p* < 0.05.

**Table 1 pharmaceutics-14-00980-t001:** Retention time, retention factor (*k*), separation factor (*α*), purity and content (μg/g) of carotenoids in sweet potato peel.

Peak No.	Compound	Retention Time (min)	Retention Factor (*k*) ^a^	Separation Factor (*α*) ^b^	Peak Purity (%)	Contents (μg/g)
1	All-trans-violaxanthin	8.69	2.03	1.46 (1, 2) ^c^	92.1	5.48
2	cis-lutein	11.17	2.38	1.07 (2, 3)	96.7	18.35
3	All-trans-lutein	11.70	2.54	1.31 (3, 4)	96.8	42.93
4	All-trans-zeaxanthin	14.32	3.34	1.54 (4, 5)	90.7	25.57
IS ^d^	All-trans-canthaxanthin	14.82	3.63	1.04 (5, IS)	99.7	-
5	15- or 15′-cis-cryptoxanthin	20.30	5.15	1.47 (IS, 6)	93.0	72.37
6	All-trans-β-cryptoxanthin	20.78	5.30	1.22 (6, 7)	94.6	155.9
7	15- or 15′-cis-β-carotene	24.56	6.44	1.03 (7, 8)	98.7	65.67
8	13- or 13′-cis-β-carotene	25.12	6.62	1.10 (8, 9)	97.7	255.8
9	All-trans-β-carotene	27.23	7.25	1.05 (9, 10)	95.6	663.8
10	All-trans-α-carotene	28.36	7.60	1.05 (9, 10)	92.3	75.59
	Total					1381.46

^a^$k=\frac{\left(t_{R}-t_{0}\right)}{t_{R}}$, where, *t_R_* and *t*_0_ are the retention times of carotenoid peak and solvent peak, respectively. ^b^$\;\alpha =\frac{t_{R2}-t_{0}}{t_{R1}-t_{0}}$, where, *t*_*R*1_ and *t*_*R*2_ are the retention times of two neighboring carotenoid peaks. ^c^ Numbers in parentheses represent two neighboring peak numbers. ^d^ IS = internal standard.

**Table 2 pharmaceutics-14-00980-t002:** Quality control data of carotenoids in sweet potato peel (TNG 66) by HPLC-DAD.

Peak No.	Carotenoids	Repeatability		Intermediate Precision		Accuracy					
		Contents (μg/g) ^a^	RSD (%) ^b^	Contents (μg/g) ^a^	RSD (%) ^b^	Original (μg)	Spiked (μg)	Found (μg)	Recovery (%) ^c^	Mean ± SD (%)	RSD (%) ^b^
1	All-trans-violanxanthin	5.17 ± 0.2	0.9	5.73 ± 1.9	1.6	-	-	-	-	-	-
2	cis-lutein	18.42 ± 0.4	2.1	18.75 ± 0.8	4.2	-	-	-	-	-	-
3	All-trans-lutein	43.21 ± 0.5	1.1	42.95 ± 1.2	2.7	45.2	5	50	96.0	94.6 ± 2.0	2.1
						45.2	50	91.8	93.2		
4	All-trans-zeaxanthin	26.13 ± 0.1	0.3	26.70 ± 0.4	1.4	27.4	5	32.1	94.0	92.1 ± 2.7	2.9
						27.4	50	72.5	90.2		
5	15- or15′-cis-β-cryptoxanthin	75.07 ± 1.1	1.4	75.51 ± 0.7	0.9	-	-	-	-	-	-
6	All-trans-β-cryptoxanthin	162.45 ± 2.5	1.5	163.02 ± 1.8	1.1	161.2	5	165.9	94.0	91.9 ± 3.0	3.2
						161.2	50	206.1	89.8		
7	15- or 15′-cis-β-carotene	71.08 ± 0.3	0.4	71.44 ± 0.4	0.5	-	-	-	-	-	-
8	13- or 13′-cis-β-carotene	258.70 ± 2.1	0.8	258.53 ± 1.8	0.6	-	-	-	-	-	-
9	All-trans-β-carotene	688.15 ± 3.2	0.4	670.28 ± 2.4	0.3	673.9	50	719.1	90.4	94.7 ± 6.0	6.4
						673.9	500	1168.5	98.9		
10	All-trans-α-carotene	77.95 ± 1.4	1.7	76.87 ± 2.0	2.6	77.3	5	81.4	82.0	86.1 ± 5.8	6.7
						77.3	50	122.4	90.2		

^a^ Data are presented as mean ± standard deviation of triplicate determinations. ^b^ RSD (%) = (standard deviation/mean) × 100. ^c^ Recovery (%) = [(amount found – original amount)/amount spiked] × 100.

**Table 3 pharmaceutics-14-00980-t003:** Average particle size, polydispersity index and zeta-potential of carotenoid nanoemulsions during storage for 90 days at 25 °C as well as both particle size and zeta potential during heating at different temperatures (40 °C, 70 °C and 100 °C).

**Storage Time (Day)**	**Particle Size (nm) ^a^**				**Polydispersity Index ^a^**		**Zeta-Potential (mV) ^a^**	
0	13.3 ± 0.4 ^B^				0.238 ± 0.01 ^B^		−69.8 ± 0.5 ^D^	
15	13.3 ± 0.3 ^B^				0.249 ± 0.01 ^AB^		−69.0 ± 0.2 ^D^	
30	13.6 ± 0.1 ^AB^				0.265 ± 0.02 ^AB^		−67.4 ± 1.0 ^C^	
45	13.9 ± 0.3 ^A^				0.267 ± 0.01 ^A^		−65.3 ± 0.8 ^B^	
60	13.5 ± 0.4 ^AB^				0.267 ± 0.02 ^A^		−66.6 ± 0.6 ^C^	
75	14.0 ± 0.2 ^A^				0.270 ± 0.03 ^A^		−64.5 ± 0.7 ^B^	
90	13.8 ± 0.4 ^AB^				0.263 ± 0.02 ^AB^		−63.0 ± 0.9 ^A^	
**Heating Time**	**Particle Size (nm) ^a^**				**Zeta Potential (mV) ^a^**			
	**0.5 h**	**1 h**	**1.5 h**	**2 h**	**0.5 h**	**1 h**	**1.5 h**	**2 h**
Control (unheated)	13.3	-	-	-	−69.0	-	-	-
40 ℃	13.4	13.9	13.6	14.0	−66.0	−62.3	−62.1	−60.1
70 ℃	14.2	14.5	15.1	15.0	−57.4	−60.1	−42.4	−41.6
100 ℃	16.9	17.5	18.0	18.2	−41.2	−40.9	−34.3	−31.2

^a^ Data shown are mean ± standard deviation (*n* = 3). Data with different letters (A–D) in the same column are significantly different at *p* < 0.05.

**Table 4 pharmaceutics-14-00980-t004:** Effect of carotenoid extracts and nanoemulsions on cell cycle distribution of human breast cancer cell MCF-7 ^1^.

Concentration (μg/mL)	Sub-G1 (%)	G0/G1 (%)	S (%)	G2/M (%)
Control	5.68 ± 0.2 ^A^	47.12 ± 0.4 ^A^	18.28 ± 0.3 ^A^	28.38 ± 0.1 ^A^
Carotenoid extract				
5	7.77 ± 0.5 ^B^	49.68 ± 0.3 ^B^	15.75 ± 0.4 ^B^	26.36 ± 0.7 ^B^
10	8.09 ± 0.2 ^B^	51.75 ± 1.2 ^C^	15.85 ± 0.4 ^B^	24.35 ± 0.5 ^C^
Carotenoid nanoemulsion				
5	8.48 ± 0.8 ^B^	53.43 ± 0.4 ^D^	15.23 ± 0.6 ^B^	22.17 ± 0.3 ^D^
10	8.09 ± 0.7 ^B^	54.65 ± 0.3 ^E^	14.68 ± 1.2 ^B^	21.97 ± 0.7 ^D^

^1^ Data shown are mean ± standard deviation (*n* = 3). Data with different letters (A–E) in the same column are significantly different at *p* < 0.05.

## Data Availability

Not applicable.

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
