# Peer review of "A Comparative Study on Inhibition of Breast Cancer Cells and Tumors in Mice by Carotenoid Extract and Nanoemulsion Prepared from Sweet Potato (Ipomoea batatas L.) Peel"

_pharmaceutics, 2022, doi:10.3390/pharmaceutics14050980_

Round 1

Reviewer 1 Report

The manuscript entitled „A Comparative Study on Inhibition of Breast Cancer Cells and Tumors in Mice by Carotenoid Extract and Nanoemulsion Prepared from Sweet Potato (Ipomoea batatas L.) Peel“ was submitted by Hsin-Yen Hsu and Bing-Huei Chen to the journal “Pharmaceutics” in order to be considered for publication as an Article. The manuscript reports on the effect of a nanoemulsion prepared from sweet potato peel on MCF-7 breast cancer cells. The comprehensive study covers the determination of the carotenoid composition of the created extract, followed by the manufacturing of a nanoemulsion. The nanoemulsion was characterized regarding characteristic parameters (zeta potential, particle size, distribution) and its stability as well as release was studied. The effect of the nanuemulsion on MCF-7 cancer cell lines was compared with the effects of the extract, while non-cancer mammary cells served for comparison. The experiments include MTT-assay, induction of caspases, flow-cytometry and finally even animal studies. All in all, the authors provide a comprehensive study. I wish to praise the very clear description of the experimental section. Nevertheless, there are some points that should be considered by the authors in order to reconsider this manuscript for publication.

Major concerns:

The MTT assay was performed at 37 °C and incubation for 48 h. The authors investigated the stability of the nanoemulsion under different conditions, however, the particular temperature of 37 °C was not considered. Therefore, it is suggested to confirm the stability of the nanoemulsion under the specific experimental conditions of the MTT assay.

The authors should provide the effect of a positive control on the cell viability of their cells. Actually, they should have adduced an established anti-cancer drug for their investigation.

The authors merely used MCF-7 breast cancer cells for their study. This cell line is known to be hormone-dependent. Apart from ER-receptors, PR-receptors and HER2-receptors also play an important role in the characterization of cancer cell lines. Lack of these receptors can be accompanied by challenge of treatment. The authors should discuss their results using literature and deduce potential conclusions on other breast cell lines, i.e. triple negative breast cancer cell lines.

Minor concerns:

“nutrients such as moisture” This statement seems strange.

“Thus, it is possible to reduce” better say: Thus, it is desirable…

“A method based on Kao et al. (2012) was modified to extract carotenoids from sweet potato peel.” Please provide the reference according to the style of the journal, i.e. using a numeral in square brackets.

“followed by adding 5 concentrations (2.5, 5, 10, 15 and 20 µg/mL) of carotenoid nanoemulsion or carotenoid extract for triplicate experiments” Was it just the nanoemulsion to be added or caretonid-containing medium?

Figure 2, subfigure A: The labelling “A” is hard to read. The authors could consider change of the color? Moreover, from which setup (concentration used to prepare the nanoemulsion) are the data of subfigures B-D from?

Figure 3: “Results are presented as mean ± standard deviation of triplicate measurements”. Is it just one experiments with triplicate measurements or even three independent experiments (which would be desireable)? Also Figure 4, Figure 5, Figure 7

Figure 5: I am wondering: Why do the authors separate the results into four subfigures. A presentation such as in Figure 4, i.e. including the data of both cell lines, would allow for easier comparison. And indeed, it would show at a first glance that the effects towards the MCF-7 cancer cell line is more pronounced.

Although there will be a close editing by the MDPI publisher, the authors are kindly asked to correct formal issues, such as consistent use of spaces with units (and not using hyphens), use of italics within chemical nomenclature, punctuation, typos (line 485: and, line 496: DMSO), etc.

Author Response

AUTHORS’ RESPONSE TO REVIEWER #1 COMMENTS FOR PHARMACEUTICS-1667444

Reviewer #1

The manuscript entitled „A Comparative Study on Inhibition of Breast Cancer Cells and Tumors in Mice by Carotenoid Extract and Nanoemulsion Prepared from Sweet Potato (Ipomoea batatas L.) Peel“ was submitted by Hsin-Yen Hsu and Bing-Huei Chen to the journal “Pharmaceutics” in order to be considered for publication as an Article. The manuscript reports on the effect of a nanoemulsion prepared from sweet potato peel on MCF-7 breast cancer cells. The comprehensive study covers the determination of the carotenoid composition of the created extract, followed by the manufacturing of a nanoemulsion. The nanoemulsion was characterized regarding characteristic parameters (zeta potential, particle size, distribution) and its stability as well as release was studied. The effect of the nanuemulsion on MCF-7 cancer cell lines was compared with the effects of the extract, while non-cancer mammary cells served for comparison. The experiments include MTT-assay, induction of caspases, flow-cytometry and finally even animal studies. All in all, the authors provide a comprehensive study. I wish to praise the very clear description of the experimental section. Nevertheless, there are some points that should be considered by the authors in order to reconsider this manuscript for publication.

Authors’ response: The authors wish to thank the reviewer for the encouraging positive comments.

Major concerns:

  1. The MTT assay was performed at 37 °C and incubation for 48 h. The authors investigated the stability of the nanoemulsion under different conditions, however, the particular temperature of 37 °C was not considered. Therefore, it is suggested to confirm the stability of the nanoemulsion under the specific experimental conditions of the MTT assay.

Authors’ response: As shown in Table 3, the nanoemulsion stability was evaluated by heating at 40, 70 and 100 °C for 2 h, and a slight change in particle size and zeta potential was shown, implying a high stability of this nanoemulsion. Also, a high stability of this nanoemulsion was found over a 90-day storage period at 25 °C. Thus, we can expect a high stability of this nanoemulsion for the MTT assay performed at 37°C and incubated for 48 h. (Please refer Table 3 in P13; L452-461).

  1. The authors should provide the effect of a positive control on the cell viability of their cells. Actually, they should have adduced an established anti-cancer drug for their investigation.

Authors’ response: I agree with the reviewer’s comment that it is important to use an anti-cancer drug for the cell culture study. However, an anti-cancer drug can be toxic to normal cells. Thus, in this study we think it is more important to use an anti-cancer drug (paclitaxel) for the animal experiment for comparison with carotenoid extract and nanoemulsion in inhibiting growth of mice tumors. 

  1. The authors merely used MCF-7 breast cancer cells for their study. This cell line is known to be hormone-dependent. Apart from ER-receptors, PR-receptors and HER2-receptors also play an important role in the characterization of cancer cell lines. Lack of these receptors can be accompanied by challenge of treatment. The authors should discuss their results using literature and deduce potential conclusions on other breast cell lines, i.e. triple negative breast cancer cell lines.

Authors’ response: A new paragraph discussing the results in our study and other breast cancer cell lines reported in the literature was provided as follows: “Clinically, breast cancer is categorized into 3 major subtypes based on the presence or absence of estrogen receptors (ER), progesterone receptors (PR), and human epidermal growth factor receptor-2 (HER2), with more than 60% of breast cancers being ER-positive and about 20% being negative for ER, PR and HER, also named as triple-negative breast cancer (TNBC) [41, 42]. As TNBC cells such as BT20, MDA468, MDA231 and MDA436 lack specific cell-surface receptors for active targeting, the treatment of TNBC remains incurable using currently available drugs. In a study dealing with metabolic alterations in TNBC cells in comparison with other subtypes of breast cancer cells using molecular and metabolic analyses, Pelicano et al. [41] demonstrated that compared to MCF-7 cells (ER/PR double positive and HER-2 negative), TNBC cells possessed special metabolic characteristics manifested by high glucose intake, increased lactate production and low mitochondrial respiration, suggesting that this metabolic intervention may be an effective therapeutic strategy for TNBC cells. Nevertheless, there are some other approaches which may be effective in curing TNBC. For instance, the development of novel drug-nanoparticles capable of accumulating in patient-derived xenograft tumors through the enhanced permeability and retention (EPR) effect in tumor vasculature may lead to new and effective TNBC treatments [43]. In other words, the preparation of carotenoid nanoemulsion from sweet potato peel in our study may be used to treat ER-, PR- or HER-2-positive tumors and TNBC through passive targeting effect.” (Please refer L701-719).

Minor concerns:

  1. “nutrients such as moisture” This statement seems strange.

Authors’ response: As suggested by the reviewer, “moisture” is now removed. (Please refer L38).

  1. “Thus, it is possible to reduce” better say: Thus, it is desirable…

Authors’ response: As suggested by the reviewer, “possible’ is now replaced with “desirable”. (Please refer L72).

  1. “A method based on Kao et al. (2012) was modified to extract carotenoids from sweet potato peel.” Please provide the reference according to the style of the journal, i.e. using a numeral in square brackets.

Authors’ response: As suggested by the reviewer, the reference “Kao et al. (2012)” is now corrected as “Kao et al. [11]. (Please refer L129, L142, L157, L165-166).

  1. “followed by adding 5 concentrations (2.5, 5, 10, 15 and 20 µg/mL) of carotenoid nanoemulsion or carotenoid extract for triplicate experiments” Was it just the nanoemulsion to be added or caretonid-containing medium?

Authors’ response: Only nanoemulsion is added. (Please refer L269-270).

  1. Figure 2, subfigure A: The labelling “A” is hard to read. The authors could consider change of the color? Moreover, from which setup (concentration used to prepare the nanoemulsion) are the data of subfigures B-D from?

Authors’ response: As suggested by the reviewer, the subfigure labelling A is now shown to be readable. Also, the Figure 2 caption is now modified to include the information as to which nanoemulsion concentration was used to obtain the data presented in subfigures B-D as follows: “Figure 2. Appearance of carotenoid nanoemulsions prepared at different concentrations: a, 1000 ppm; b, 3000 ppm, c, 5000 ppm, d, 8000 ppm, e, 10000 ppm (A) along with particle size distribution as determined by a dynamic light scattering method (B) as well as their TEM image (C) and associated particle size distribution (D) from 10000 ppm of carotenoid nanoemulsion.” (Please refer Figure 2 in P12)

  1. Figure 3: “Results are presented as mean ± standard deviation of triplicate measurements”. Is it just one experiments with triplicate measurements or even three independent experiments (which would be desireable)? Also Figure 4, Figure 5, Figure 7

Authors’ response: Actually Figure 3 is one experiment with triplicate measurements, while Figures 4, 5 and 7 are triplicate independent experiments. Appropriate modifications are now made in the captions of respective figures. (Please refer L484, L512, L548, L633-634).

  1. Figure 5: I am wondering: Why do the authors separate the results into four subfigures. A presentation such as in Figure 4, i.e. including the data of both cell lines, would allow for easier comparison. And indeed, it would show at a first glance that the effects towards the MCF-7 cancer cell line is more pronounced.

Authors’ response: As suggested by the reviewer, the Figure 5 is redrawn to combine two carotenoid nanoemulsion subfigures into one and likewise two carotenoid extract subfigures into one. (Please refer Figure 5 in P16).

  1. Although there will be a close editing by the MDPI publisher, the authors are kindly asked to correct formal issues, such as consistent use of spaces with units (and not using hyphens), use of italics within chemical nomenclature, punctuation, typos (line 485: and, line 496: DMSO), etc.

Authors’ response: As suggested by the reviewer, the use of hyphens is now replaced with spaces between a value and its unit wherever appropriate. Also, the italics within chemical nomenclature, punctuation and indicated typos are corrected. (Please refer L64, L65, L122, L217, L226, L239, L240, L246, L262, L264, L268, L270, L280-281, L284, L293, L323, L337 & 538-539 as well as L491-492 & L502).

Reviewer 2 Report

Question 1: In your description for cell culture experiment under method, what do you mean by “cell sap”? Can you also clarify how you culture cells with 7% DMSO? Generally, any percentage higher than 1% have toxic effect on cells.

Question 2: In assessment of the vehicle controls in cell culture study, did you incubate the same amount of time (48 hrs) as your subsequent treatment? You did not have any information in that assessment.

Question 3:  In your cell cycle and apoptosis study, have you also explore the impact of the carotenoids on the MCF10 as well as the extract also has an impact on their viability?

Question 4: In your flow cytometry figure, can you add the percentage in each quadrant of the flow chart?

Question 5: By looking at your scale in your FlowJo scatter plots, you have inconsistent gating ( Y-axis in control vs treatment). This suggests that data may be of different run. Can you clarify this difference?

Question 6: In your cell cycle and apoptosis assay, do you have a positive control that is known to cause a block in cell cycle and apoptosis respectively, to better demonstrate the impact of your test product?

Question 7: Given that you have conducted the molecular assays on the cell lines, have you examined whether you will see the same effect (apoptosis and cell cycle changes) on the tumour tissues you have collected from your mouse model?

Question 8: MCF10 may be a "normal" cells but it is still a cell line. Have the authors have access to normal epithelial cells and assess the effect of carotenoids on them?

Author Response

AUTHORS’ RESPONSE TO REVIEWER #2 COMMENTS FOR PHARMACEUTICS-1667444

Reviewer #2

  1. Question 1: In your description for cell culture experiment under method, what do you mean by “cell sap”? Can you also clarify how you culture cells with 7% DMSO? Generally, any percentage higher than 1% have toxic effect on cells.

Authors’ response: “cell sap” means “cell juice” or “cytosol”. To make it clear “cell sap” is now corrected as “cytosol”. DMSO can be used as an anti-freezing agent to control formation of ice crystal size and prevent cell injury during freezing of cells. Following fast thawing, one mL cytosol was transferred to a culture plate and 8-mL fresh culture medium was added for cell attachment in an incubator. We just followed the standard protocol of Taiwan Bioresource Collection and Research Center. But for the subsequent cell culture experiment, the DMSO dose was controlled at 2% to maintain high viability of both MCF-7 and HG184B5F5/M10 cells. (Please refer L254; L251-262).

  1. Question 2: In assessment of the vehicle controls in cell culture study, did you incubate the same amount of time (48 hrs) as your subsequent treatment? You did not have any information in that assessment.

Authors’ response: The group without sample treatment was used a control and was incubated for 48 h. This information is now included in the text. (Please refer L274-275).

  1. Question 3:  In your cell cycle and apoptosis study, have you also explore the impact of the carotenoids on the MCF10 as well as the extract also has an impact on their viability?

Authors’ response: In this study we focus on inhibition mechanism of human breast cancer cells MCF-7 and mice tumors. Thus, in the cell cycle and apoptosis study, we only explore the effect of carotenoid extract and nanoemulsion on MCF-7 cells. Furthermore, based on the result of MTT assay, a high cell viability of human breast epithelial cell line H184B5F5/M10 was maintained following treatment of carotenoid extract and nanoemulsion at 5 and 10 mg/mL. (Please refer L276-302).

  1. Question 4: In your flow cytometry figure, can you add the percentage in each quadrant of the flow chart?

Authors’ response: As suggested by the reviewer, the percentage is added in each quadrant of flow cytometry Figure 6. (Please refer Figure 6 in P17).

  1. Question 5: By looking at your scale in your FlowJo scatter plots, you have inconsistent gating ( Y-axis in control vs treatment). This suggests that data may be of different run. Can you clarify this difference?

Authors’ response: As suggested by reviewer, the control figure obtained by the same run is now provided with the y-axis scale similar to that shown in other treatment figures. (Please refer Figure 6 in P17).

  1. Question 6: In your cell cycle and apoptosis assay, do you have a positive control that is known to cause a block in cell cycle and apoptosis respectively, to better demonstrate the impact of your test product?

Authors’ response: For positive control, we studied the effect of 1% sodium azide and 500 mM H2O2 on cell cycle and apoptosis of MCF-7 cells. Please refer to the figures (please see the attachment Word file for figures). However, the data is not shown in the text as we thought positive control is only used as a verification treatment of experimental procedure. Instead, we thought it is more important to use an anti-cancer drug (paclitaxel) in the animal experiment for comparison of anti-tumor effect with carotenoid extract and nanoemulsion prepared from sweet potato peel.

  1. Question 7: Given that you have conducted the molecular assays on the cell lines, have you examined whether you will see the same effect (apoptosis and cell cycle changes) on the tumour tissues you have collected from your mouse model?

Authors’ response: In this study we only measured tumor size and volume as well as EGF and VEGF in serum. However, for future study, it is important to examine tumor tissue to see if the same effect of apoptosis and cell cycle is shown.

  1. Question 8: MCF10 may be a "normal" cells but it is still a cell line. Have the authors have access to normal epithelial cells and assess the effect of carotenoids on them?

Authors’ response: In this study human breast epithelial cell line H184B5F5/M10 is the only human mammary epithelial cell we can obtain from Taiwan Bioresource Collection and Research Center. In a previous study we also demonstrated that a high cell viability of fibroblast cells CCD986SK was maintained following treatment of carotenoid extract and nanoemulsion prepared from pomelo leaves at 5 and 10 mg/mL. Please refer to our recently published research article: Plants 2021, 10, 2129 (https://doi.org/10.3390/ plants10102129). (Please refer L116-118).

Round 2

Reviewer 1 Report

The authors Hsin-Yen Hsu and Bing-Huei Chen have submitted a revised version of their manuscript “A Comparative Study on Inhibition of Breast Cancer Cells and Tumors in Mice by Carotenoid Extract and Nanoemulsion Prepared from Sweet Potato (Ipomoea batatas L.) Peel.”

The authors made every effort to incorporate all of the concerns and suggestions previously addressed as they revised their manuscript. They have corrected the errors mentioned, made additions, or provided reasonable rebuttals. Overall, I suggest a further processing of the manuscript. All the best!

Author Response

AUTHORS’ RESPONSE TO REVIEWER # 1 COMMENTS FOR PHARMACEUTICS-1667444

The authors Hsin-Yen Hsu and Bing-Huei Chen have submitted a revised version of their manuscript “A Comparative Study on Inhibition of Breast Cancer Cells and Tumors in Mice by Carotenoid Extract and Nanoemulsion Prepared from Sweet Potato (Ipomoea batatas L.) Peel.”

The authors made every effort to incorporate all of the concerns and suggestions previously addressed as they revised their manuscript. They have corrected the errors mentioned, made additions, or provided reasonable rebuttals. Overall, I suggest a further processing of the manuscript. All the best!

Author’s response: The authors thank the Reviewer #1 for accepting our responses to all his comments and recommending this research article for publication in Pharmaceutics.

Reviewer 2 Report

Thank you for your response. Just one last comment, I would recommend to include your comments about future work in your discussion. 

Author Response

AUTHORS’ RESPONSE TO REVIEWER # 2 COMMENTS FOR PHARMACEUTICS-1667444

 Reviewer #2

Thank you for your response. Just one last comment, I would recommend to include your comments about future work in your discussion. 

Author’s response: As suggested by the Reviewer #2, we have now included the future work in the last two sentences of results and discussion as “However, in this study we only measured tumor size and volume as well as EGF and VEGF in mice serum. Thus, for future study, it is important to examine tumor tissue to find out if the same effect of apoptosis and cell cycle is shown.”

The authors also thank the Reviewer #2 for accepting our responses to all his comments for a possible publication of this research article in Pharmaceutics.